# A cross-sectional study of dental students perception of dental faculty gender differences

**Shaista Rashid** [1] *, **Mohamed ElSalhy** [2]

**1** A.T. Still University - Missouri School of Dentistry & Oral Health, St Louis, Missouri, United States of America, **2** College of Dental Medicine, University of New England, Portland, Maine, United States of America

* shaistarashid@atsu.edu

**Data Availability Statement:** Due to sensitive nature of the topic and small group of participants there is a risk of indirect identification of study participants. Data will be available upon request

## Abstract

### Objective

The objective of this study was to evaluate students' perceptions of differences in learning from faculty of different gender.

### Method

This cross-sectional study involved pre-doctoral dental students (years 2 to 4) who had a simulation and/or clinical experience working with dental faculty for at least one year. Students completed a self-administered questionnaire with three sections: demographic, difference between faculty related to their knowledge, skill, critical thinking, acceptance of cultural differences, and students' preferences in working with faculty in specialty clinics.

### Results

A total of 136 students completed the survey (75.4% response rate). Participants were 52.6% women, 62.2% self-identified as Caucasian/White. Students reported that female faculty are more understanding (p = 0.001) and accepting of cultural differences (p<0.001) compared to male faculty (p<0.05). Students reported perceiving female faculty more as being a role model than male faculty (p = 0.034). When comparing male and female students, male student's perception of male faculty as a role model was significantly higher than female students (p<0.05). There was no significant difference in student's perceptions between male and female faculty in their knowledge, skills, compassion, critical thinking, providing feedback, communication skills, and grading (p>0.05). Caucasian/White students perceived female faculty as more encouraging for discussions and male faculty as more rigid/inflexible (p<0.05).

### Conclusions

Students perceived female faculty as more understanding and culturally competent compared to male faculty. There were no significant differences in student's perceptions of male

from Julie Peterson Chair of the UNE Institutional Review Board at (207) 221-4567 or irb@une.edu.

**Funding:** The authors received no funding for this work.

**Competing interests:** The authors have declared that no competing interests exist.

and female faculty in their knowledge, skills, compassion, critical thinking, feedback, communication skills, and grading. Students perceived female faculty as role models more than male faculty.

## Introduction

The gender composition of the dental profession in the US has changed over the past two decades, with more females entering the field of dentistry. In 1978, only 11.2% of dental students were female compared to 50.5% in 2018–19 [1]. Female dentists accounted for 16% of working dentists in 2001, which increased to 33.4% in 2019 [2]. Although the number of female dentists increased, such an increase has not been reflected among dental faculty. According to the American Dental Education Association, the number of females in 2019 accounts for 40.5% of full-time and 35% of part-time dental faculty positions, with the highest rates in the Midwest regions [3]. Although women are more likely than men to choose academic careers, female dentists remain underrepresented in the academic workforce of the United States [4–6] It has been well-documented in medicine that mentors and role models can have an impact on different genders' choices of a specific specialty over another [7, 8]. Gender disparity in choosing dental specialties and leadership roles is due to a lack of sufficient women role models and mentorship for young female dentists [9, 10]. The increase in female student enrollment in dentistry requires gender diversification of dental educators to provide mentorship, optimal learning environment, assorted perspective in career advice early in education, and guidance in choosing the specialties [11, 12]. Female educators can support young dental professionals' growth and help them shape career choices, clinical attributes, and professional qualities through motivation, inspiration, and example [11, 13].

A substantial amount of research in the medical field has documented the difference in teaching styles based on gender. Studies done with medical students have reported that men perform better in terms of substantial knowledge and procedural skills while women have more robust communication and diagnostic skills [5, 14]. Other studies have found that women have a more professional attitude and provide better feedback while men are more knowledgeable [15, 16].

The learning environment in dental school includes teaching in the classroom, simulation clinic, and clinical setting. Clinical and simulation lab teaching is based on one-to-one interaction for extended hours where educators provide professional and personal guidance in the context of patient care [17]. Although differences in faculty interaction based on gender have been evaluated in medical education, such differences in dental education have not been examined. Therefore, this study aimed to evaluate students' perceptions of differences in learning from faculty of a different gender.

## Method

### Study setting and participation

Participants in this cross-sectional study were second, third, and final year pre-doctoral dental students at the University of New England College of Dental Medicine. All students were invited to participate. All participants received a cover letter and verbal briefing explaining the content of the study. For student privacy protection, no identifiable information was included in the survey. The entire class was presented with the survey and was informed not to fill it out

if they did not wish to participate. All surveys were collected back regardless of the participation. No signed consent was obtained. Filling out the survey was used as implied informed consent. The Protocol was approved by the University of New England Institutional Review Board (Protocol No. 19.10.04–008).

### Data collection

Data were collected using a self-administered questionnaire adopted from the cognitive apprenticeship model in clinical practice [18]. Three faculty experts in public health, gender studies, and education evaluated the survey for content validity. After content validity, the survey was pre-tested with eight students of different age, gender, race and school year. A debriefing session was held following the pre-test completion to detect ambiguity of words, misinterpretation of answering scales, inability to answer sensitive questions, or any other problems with the questionnaire. The questions were adjusted based on participants' comments. Students involved in the validation process were asked not to participate in the survey. The questionnaire (S1 File) consisted of three sections. The first section collected demographic information including age, gender, ethnicity, and current year of dental school. The second section consisted of twenty-two questions exploring gender-related perceptions. It included four questions about the difference in knowledge, six questions about communication and teaching skills, ten questions about grading and feedback, and two questions regarding cultural sensitivity. The third section inquired about student's preferences for working with specific specialty based on the gender of the faculty. Students were asked to score faculty of different gender using a five point scale with five being the highest score. All questions included a not applicable (N/A) option.

### Data analysis

Data were managed and analyzed using SPSS 21.0 software (IBM Corp., Armonk, N.Y., USA). Data were tested for normality using the Shapiro–Wilk test. As data were normally distributed, means and standard deviations were used for data description. Difference in perception scores according to student's demographics were evaluated using an independent-sample t-test. The level of significance was set at 0.05.

## Results

A total of 136 students completed the survey (75.4% response rate). Participants were 52.6% females, 62.2% self-identified as Caucasian/White and 54.5% were under the age of 26 years (Table 1). Most students have worked with both male and female faculty for at least one session per week in both the simulation clinic and during patient care (Table 2).

Students reported that female faculty are more understanding (p = 0.001) and accepting of cultural differences compared to male faculty (p<0.001). Students also reported that they perceive female faculty more as being a role model than male faculty (p = 0.034) (Table 3). When comparing gender, male student's perception of male faculty as a role model was significantly higher than female students (p<0.05) with no difference between male and female students in preserving female faculty as role models (p>0.05) (Table 4).

Students did not perceive a significant difference among women and men faculty in their knowledge, skills, compassion, critical thinking, providing feedback, communication skills, and grading (p>0.05) (Table 3). Caucasian/White students perceived women faculty as more encouraging for discussions and male faculty as more rigid/inflexible (p<0.05). There was no significant difference in students perception scores according to their age or year of study

**Table 1. Participants' demographics.**

| Demographic Characteristics | N | % |
|---|---|---|
| **Year** | | |
| D2 | 54 | 39.7 |
| D3 | 54 | 39.7 |
| D4 | 28 | 20.6 |
| **Age** | | |
| 20–23 | 5 | 3.7 |
| 24–26 | 68 | 50.7 |
| 27–30 | 40 | 29.9 |
| Greater than 30 | 21 | 15.7 |
| **Gender** | | |
| Male | 64 | 47.4 |
| Female | 71 | 52.6 |
| **Ethnicity** | | |
| Caucasian/White | 84 | 62.2 |
| African American/Black | 4 | 3.0 |
| Hispanic/Latino | 12 | 8.9 |
| Asian | 21 | 15.6 |
| Mixed/multiple ethnic groups | 7 | 5.2 |

(p>0.05). Student's perceptions of differences between male and female faculty according to student's gender and ethnicity are shown in Table 4.

Students reported no preference in working with male or female faculty in both the simulation clinic and patient care (p>0.05). The gender or the ethnicity of the students did not impact their preference of working with a male or female faculty (p>0.05). Students also reported no preference in working with male or female faculty from different specialties except pediatric dentistry. Students preferred working with a female faculty while performing pediatric procedures (Table 5). When comparing student's preferences based on student's gender and ethnicity, Caucasian/White students have more preferences to work with male faculty for restorative and periodontal procedures compared to students from different ethnicities. There was no significant difference in students preferences scores according to their age or year of study (p>0.05). Student's preferences of working with male or female faculty according to student's gender and ethnicity are shown in Table 6.

## Discussion

Students perceived both male and female faculty similarly in their knowledge, skills, compassion, critical thinking, feedback, communication skills, and grading. Female faculty were more understanding and accepting of cultural differences. Students perceived female faculty as more

**Table 2. Student's exposure to faculty of different genders.**

| Sessions per week | Female Faculty- Simulation Clinic | | Female Faculty- Clinic | | Male Faculty- Simulation Clinic | | Male Faculty- Clinic | |
|---|---|---|---|---|---|---|---|---|
| | N | % | N | % | N | % | N | % |
| **Never** | 10 | 8.2 | 16 | 11.9 | 8 | 6.6 | 1 | 0.8 |
| **1–2** | 48 | 39.3 | 61 | 45.5 | 19 | 15.6 | 24 | 18.0 |
| **3 or more** | 64 | 52.5 | 57 | 42.5 | 95 | 77.9 | 108 | 81.2 |

**Table 3. Student's mean (SD) perceptions scores for male and female faculty.**

| Items | Male Faculty | Female Faculty | P-value* |
|---|---|---|---|
| **Provides constructive feedback** | 4.61 (.762) | 4.69 (.667) | 0.358 |
| **Approachable** | 4.58 (.725) | 4.61 (.747) | 0.737 |
| **Knowledgeable/skills** | 4.75 (.617) | 4.81 (.554) | 0.400 |
| **Encourages Critical thinking** | 4.64 (.690) | 4.70 (.620) | 0.451 |
| **Encourages discussions** | 4.60 (.752) | 4.74 (.593) | 0.089 |
| **Work effectively with others** | 4.580 (.755) | 4.720 (.582) | 0.134 |
| **Supportive** | 4.63 (.654) | 4.72 (.658) | 0.259 |
| **Motivational/Encouraging** | 4.53 (.842) | 4.70 (.620) | 0.096 |
| **Communicates well with patients** | 4.61 (.769) | 4.71 (.606) | 0.235 |
| **Communicates well with students** | 4.57 (.705) | 4.71 (.583) | 0.117 |
| **Communicates well with staff** | 4.67 (.680) | 4.77 (.508) | 0171 |
| **Works effectively with other faculty** | 4.54 (.751) | 4.68 (.632) | 0.097 |
| **Is a role model** | 4.48 (.995) | 4.73 (.657) | 0.034 |
| **More accepting to cultural differences** | 4.29 (.996) | 4.74 (.591) | <0.001 |
| **Provides better understanding of cultures** | 4.30 (1.062) | 4.73 (.651) | 0.001 |
| **Trust your clinical judgment more** | 4.55 (.848) | 4.46 (.834) | 0.378 |
| **Gives more independence in performing procedures** | 4.60 (.859) | 4.48 (.780) | 0.297 |
| **Challenges your knowledge/credibility** | 4.53 (.814) | 4.66 (.691) | 0.157 |
| **More rigid/inflexible** | 4.12 (1.178) | 4.26 (1.103) | 0.313 |
| **Exerts superiority** | 3.96 (1.231) | 4.10 (1.221) | 0.347 |
| **Harder on grading** | 4.12 (1.332) | 3.80 (1.375) | 0.120 |

*Independent-sample t-test

understanding and culturally competent than male faculty. Both male and female students perceived female faculty as role models, while only male students perceived male faculty as role models.

Implicit gender bias has been reported in students' medical and non-medical academic faculty evaluations [19–23]. One study reported that women are evaluated differently from men in two key areas [24]. First, women are evaluated on different criteria, including their personality, appearance, competency, and perception of intelligence, than men. Second, even when women are teaching identical courses and factors like personality and appearance are held constant, they are rated more poorly than men [24]. Previous reports from non-dental fields showed that students perceived male faculty as agentic, assertive, competent, and having higher leadership skills. In comparison, female faculty are perceived as communal types with better interpersonal and communication skills [21, 22, 24, 25]. Our study found no difference in student perception of clinical knowledge, critical thinking, and personality traits of communication, approachability, motivation, trust, and feedback based on the gender of the faculty. Additionally, such equal perception was seen regardless of the student's gender and/or ethnicity. As dental students spend large amounts of one-to-one time with dental faculty, their responses are based on real experiences rather than pure perceptions, which may control some of their biases resulting in an equal evaluation of both genders. Even though the dental profession has a patriarchal past, students' perceptions in the study show that dental education has a particularity that promotes gender equality.

A wealth of studies have reported systemic biases in faculty expectations for students' education attainment [26–28]. Faculty expectation directly impacts their investment and engagement with students. Our study reported that Caucasian/White students compared to minority

**Table 4. Student's mean (SD) perception scores for male and female faculty according to student's gender and ethnicity.**

| Items | Male Faculty | | Female Faculty | | Male Faculty | | Female Faculty | |
|---|---|---|---|---|---|---|---|---|
| | Male | Female | Male | Female | White | Others | White | Others |
| **Provides constructive feedback** | 4.62 (.855) | 4.59 (.682) | 4.68 (.713) | 4.69 (.635) | 4.69 (.620) | 4.49 (.920) | 4.67 (.752) | 4.71 (.549) |
| **Approachable** | 4.66 (.717) | 4.51 (.735) | 4.74 (.664) | 4.49 (.805) | 4.62 (.637) | 4.52 (.836) | 4.70 (.558) | 4.48 (.937) |
| **Knowledgeable/skills** | 4.70 (.707) | 4.80 (.528) | 4.72 (.701) | 4.89 (.369) | 4.75 (.571) | 4.75 (.686) | 4.73 (.660) | 4.91 (.358) |
| **Encourages Critical thinking** | 4.72 (.607) | 4.56 (.756) | 4.64 (.663) | 4.75 (.584) | 4.71 (.584) | 4.53 (.815) | 4.77 (.560) | 4.59 (.693) |
| **Encourages discussions** | 4.63 (.727) | 4.57 (.783) | 4.78 (.587) | 4.70 (.607) | 4.66 (.680) | 4.52 (.849) | 4.85 (.444)* | 4.57 (.737) |
| **Work effectively with others** | 4.56 (.787) | 4.58 (.738) | 4.72 (.573) | 4.72 (.601) | 4.62 (.687) | 4.50 (.849) | 4.75 (.541) | 4.67 (.644) |
| **Supportive** | 4.63 (.668) | 4.62 (.652) | 4.65 (.805) | 4.78 (.498) | 4.68 (.566) | 4.55 (.772) | 4.77 (.560) | 4.65 (.783) |
| **Motivational/Encouraging** | 4.52 (.839) | 4.53 (.858) | 4.70 (.580) | 4.71 (.658) | 4.65 (.652) | 4.33 (1.052) | 4.79 (.517) | 4.56 (.734) |
| **Communicates well with patients** | 4.67 (.781) | 4.55 (.765) | 4.75 (.565) | 4.67 (.648) | 4.69 (.593) | 4.48 (.969) | 4.73 (.582) | 4.68 (.650) |
| **Communicates well with students** | 4.60 (.728) | 4.54 (.693) | 4.64 (.663) | 4.77 (.504) | 4.65 (.575) | 4.45 (.861) | 4.73 (.574) | 4.67 (.606) |
| **Communicates well with staff** | 4.67 (.694) | 4.66 (.678) | 4.71 (.582) | 4.83 (.430) | 4.73 (.582) | 4.57 (.801) | 4.76 (.536) | 4.78 (.475) |
| **Works effectively with other faculty** | 4.64 (.631) | 4.44 (.850) | 4.68 (.587) | 4.67 (.683) | 4.57 (.745) | 4.50 (.773) | 4.69 (.620) | 4.65 (.662) |
| **Is a role model** | 4.78 (.511)* | 4.20 (1.172) | 4.69 (.619) | 4.76 (.699) | 4.57 (.846) | 4.33 (1.097) | 4.77 (.563) | 4.67 (.778) |
| **More accepting to cultural differences** | 4.42 (.906) | 4.15 (1.073) | 4.72 (.573) | 4.77 (.609) | 4.37 (.863) | 4.17 (1.167) | 4.77 (.533) | 4.70 (.674) |
| **Provides better understanding of cultures** | 4.41 (1.024) | 4.18 (1.101) | 4.65 (.737) | 4.80 (.566) | 4.44 (.861) | 4.10 (1.265) | 4.80 (.595) | 4.65 (.720) |
| **Trust your clinical judgment more** | 4.65 (.663) | 4.43 (1.000) | 4.47 (.819) | 4.44 (.861) | 4.62 (.721) | 4.43 (1.010) | 4.47 (.842) | 4.44 (.838) |
| **Gives more independence in performing procedures** | 4.67 (.689) | 4.52 (1.000) | 4.47 (.793) | 4.48 (.779) | 4.63 (.828) | 4.55 (.916) | 4.43 (.840) | 4.53 (.702) |
| **Challenges your knowledge/credibility** | 4.58 (.821) | 4.48 (.818) | 4.69 (.624) | 4.64 (.754) | 4.56 (.772) | 4.49 (.883) | 4.71 (.671) | 4.59 (.726) |
| **More rigid/inflexible** | 4.13 (1.191) | 4.10 (1.182) | 4.16 (1.161) | 4.35 (1.052) | 4.20 (1.102)* | 4.00 (1.285) | 4.42 (.944) | 4.02 (1.275) |
| **Exerts superiority** | 4.07 (1.223) | 3.83 (1.243) | 4.14 (1.207) | 4.04 (1.250) | 4.21 (1.081) | 3.58 (1.348) | 4.21 (1.109) | 3.92 (1.363) |
| **Harder on grading** | 4.25 (1.214) | 3.98 (1.438) | 4.02 (1.334) | 3.59 (1.408) | 4.20 (1.354) | 3.97 (1.320) | 3.92 (1.368) | 3.61 (1.386) |

* p < 0.05 (Independent-sample t-test)

students perceived male faculty as more rigid in grading. These results align with earlier studies in non-medical fields where faculty showed lower expectations and potential for educational excellence from minority students than Caucasian/White students [26–28].

Cultural competence is integral to health literacy. The literature shows that both dentists and dental students lack the skills necessary to interact in cross-cultural environments and comprehensively treat patients from different cultures [29]. Faculty attitudes and behaviors regarding cultural awareness can help shape students' cultural diversity. Previous studies have reported that clinical faculty are generally culturally diverse with no differences based on gender [30, 31]. In contrast, this study showed that students found female faculty more accepting

**Table 5. Student's mean (SD) preference scores of working with male or female faculty in different specialty areas.**

| Items | Male Faculty | Female Faculty | P-Value* |
|---|---|---|---|
| **Restorative procedures** | 4.510 (.899) | 4.640 (.707) | 0.342 |
| **Oral Surgery procedures** | 4.630 (.858) | 4.380 (1.081) | 0.219 |
| **Pediatric procedures** | 3.96 (1.149) | 4.78 (.573) | <0.001 |
| **Orthodontic procedures** | 4.50 (.916) | 4.67 (.736) | 0.093 |
| **Endodontics procedures** | 4.48 (.892) | 4.38 (1.173) | 0.429 |
| **Periodontal procedures** | 4.47 (.905) | 4.61 (.828) | 0.184 |

*Independent-sample t-test

**Table 6. Student's mean (SD) preference scores of working with male or female faculty according to student's gender and ethnicity.**

| Items | Male Faculty | | Female Faculty | | Male Faculty | | Female Faculty | |
|---|---|---|---|---|---|---|---|---|
| | Male | Female | Male | Female | White | Others | White | Others |
| **Restorative procedures** | 4.52 (1.029) | 4.50 (.804) | 4.82 (.476) | 4.51 (.810) | 4.74 (.612)* | 4.11 (1.155) | 4.68 (.708) | 4.56 (.712) |
| **Oral Surgery procedures** | 4.81 (.557) | 4.47 (1.042) | 4.33 (1.188) | 4.42 (1.018) | 4.76 (.760) | 4.40 (.995) | 4.33 (1.209) | 4.47 (.834) |
| **Pediatric procedures** | 3.96 (1.301) | 3.97 (1.045) | 4.76 (.577) | 4.79 (.577) | 4.05 (1.177) | 3.80 (1.105) | 4.79 (.559) | 4.75 (.608) |
| **Orthodontic procedures** | 4.85 (.555) | 4.26 (1.046) | 4.92 (.277) | 4.50 (.889) | 4.67 (.840) | 4.29 (.994) | 4.72 (.669) | 4.60 (.828) |
| **Endodontics procedures** | 4.69 (.736) | 4.32 (.976) | 4.68 (.945) | 4.09 (1.311) | 4.59 (.865) | 4.30 (.926) | 4.63 (1.013) | 4.10 (1.300) |
| **Periodontal procedures** | 4.58 (.881) | 4.35 (.935) | 4.65 (.755) | 4.58 (.902) | 4.77 (.710)* | 4.10 (.995) | 4.74 (.677) | 4.40 (1.000) |

* $p < 0.05$ (Independent-sample t-test)

and understanding of cultural differences than their male counterparts. This difference could also be due (sic) to sex stereotyping of women as nurturing, kind, sympathetic, and understanding. If put in the context of diversity, these traits enhance cultural competencies [32, 33]. Another potential explanation is that the majority of female faculty in the college are either from a minority background and/or have significant experience working with patients from different backgrounds, which makes them more culturally diverse.

Mentors and role models are essential for dental students' success. They provide constructive feedback and help students adapt to the challenges and changes of a new profession to overcome ambiguity and self-doubt. Studies have shown that graduates with encouraging mentors were twice more engaged at work and had better job satisfaction and leadership qualities [34, 35]. In this study, students perceived both men and women as motivational, supportive, and encouraging for critical thinking—students from both genders perceived female faculty as role models. However, only male students saw a role model in the male faculty. Similar results were seen among undergraduate students [36]. The same gender preference can explain the results. Since academic dentistry is still a male-dominated field, male students can imagine themselves in these roles compared to female students [3]. We postulate that this difference highlights the importance of gender matching and having females in academic roles to inspire female students.

In addition, students' decision to choose a postgraduate specialty can depend on encouraging mentors and exposure to gender-diverse faculty in different specialties [37]. In our study, male or female students showed no gender preference in working with any faculty from most specialties. Students perceived both men and women in different specialties as equally motivational, supportive, and encouraging of critical thinking and advanced training. However, this may not reflect their decision to pursue a specific dental specialty. Part-time faculty represented 50.3% of U.S dental school faculty in 2018–2019 [3]. These faculty engage in private practice and are more likely to endorse self-employment practices than postgraduate training [11]. It has been reported that females are underrepresented in specialties like oral surgery and endodontics, with more preferences toward pediatric dentistry, oral medicine, and general dentistry [11]. The lack of female role models and mentors has been reported as a factor in the internal segregation of dental specialties [38]. It will be interesting to investigate how exposure to different specialists from different genders influences students' decision to pursue a particular specialty in future studies.

One of the limitations of this study is that it involved students from only one school. Different schools may have different student and faculty compositions, affecting overall student perceptions. Another limitation is that only differences based on gender were evaluated. Students' perceptions can be influenced by the faculty's ethnicity as well as their educational

background. In this study, all non-White students were grouped into the "Others" category due to the small size of participants from each population. There are likely differences with members of this groups in their perceptions and experiences. Given these limitations, generalizability related to all non-White students is limited.

## Conclusion

In conclusion, students' perceptions can be colored by gender expectations, which influence their learning experiences. To attain gender diversification, dental schools should improve younger female faculty recruitment and support their advancement in academia and leadership roles. The importance of role models and mentors cannot be overemphasized. Female faculty in dental schools and private practices can be potential mentors for younger or less experienced female dentists. To increase dental faculty cultural diversity, schools need to provide faculty development seminars on gender and cultural awareness. Since this study was conducted only at one dental school, studies about student differences in perception based on gender in other dental schools would be valuable.

## Supporting information

**S1 File.**
(DOCX)

## Author Contributions

**Conceptualization:** Shaista Rashid, Mohamed ElSalhy.

**Data curation:** Shaista Rashid.

**Investigation:** Shaista Rashid, Mohamed ElSalhy.

**Methodology:** Shaista Rashid, Mohamed ElSalhy.

**Project administration:** Shaista Rashid.

**Validation:** Mohamed ElSalhy.

**Writing – original draft:** Shaista Rashid.

**Writing – review & editing:** Mohamed ElSalhy.

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
