## [Decision Letter · Decision Letter 0]

6 Apr 2022

PONE-D-21-33779Dental Student’s Perception of Gender Differences among Dental Faculty: A Cross-sectional StudyPLOS ONE

Dear Dr. Shaista Rashid,

Thank you for submitting your manuscript to PLOS ONE. After careful consideration, we feel that it has merit but does not fully meet PLOS ONE’s publication criteria as it currently stands. Therefore, we invite you to submit a revised version of the manuscript that addresses the points raised during the review process.

We look forward to receiving your revised manuscript.

Kind regards,

Carlos Miguel Rios-González, Ph.D

Academic Editor

PLOS ONE

https://journals.plos.org/plosone/s/file?id=ba62/PLOSOne_formatting_sample_title_authors_affiliations.pdf".

2. Please change "female” or "male" to "woman” or "man" as appropriate, when used as a noun (see for instance https://apastyle.apa.org/style-grammar-guidelines/bias-free-language/gender).

4. Please provide additional details regarding participant consent. In the ethics statement in the Methods and online submission information, please ensure that you have specified what type you obtained (for instance, written or verbal, and if verbal, how it was documented and witnessed). If your study included minors, state whether you obtained consent from parents or guardians. If the need for consent was waived by the ethics committee, please include this information.

Reviewers' comments:

Reviewer's Responses to Questions

**Comments to the Author**

1. Is the manuscript technically sound, and do the data support the conclusions?

Reviewer #1: Yes

Reviewer #2: Yes

Reviewer #3: Partly

2. Has the statistical analysis been performed appropriately and rigorously? 

Reviewer #1: Yes

Reviewer #2: Yes

Reviewer #3: I Don't Know

3. Have the authors made all data underlying the findings in their manuscript fully available?

Reviewer #1: Yes

Reviewer #2: Yes

Reviewer #3: No

4. Is the manuscript presented in an intelligible fashion and written in standard English?

Reviewer #1: Yes

Reviewer #2: Yes

Reviewer #3: No

5. Review Comments to the Author

Reviewer #1: This is a very interesting study. Please note this reviewer’s comments below.

Title:

Regarding the title of Dental Student’s Perception of Gender Differences among Dental Faculty: A Cross- sectional Study

It might be better stated as the following since it only relates to one dental school approach or as: A Cross-sectional Study of Dental Student Perception of Dental Faculty Gender Differences

Throughout the manuscript, please check for clarity and sentence structure.

Page 3 Lines 56-57 regarding

Mentors and role models’ effect on gender related specialty preference has been well-

documented in medicine. [8,9]

Please rephrase this statement for clarity.

Page 4 Lines 75-76 regarding

Survey was administered separately among all clinical years in class and collected from all students regardless of participation.

Can you please explain what is meant by regardless of participation?

Line 77 regarding

No signed consent forms were collected to protect student privacy. Filling the survey was used as implied informed consent.

Please rephrase these two sentences for clarity.

Page 6 Table 1, is this table necessary since the content has been documented within the manuscript.

Page 13 Line 173 regarding

This difference highlights the importance of gender matching and having females in academic roles to inspire female students.

Do you have a reference for this sentence?

Page 14 Line 192 regarding

To increase dental faculty cultural competency, schools need to provide faculty development seminars on gender and cultural awareness.

Did you mean cultural diversity vs. cultural competency? One can never be fully culturally competent since it is a journey. The word cultural competency shows up during the entire manuscript whereas a better fit can be cultural diversity and not competency.

Regarding Methodology

Can you please share the survey as a table so that others can duplicate this work?

Can you explain the high survey response rate if it was on a volunteer basis and confidential?

Did you have a chance to compare D2, to D3 and D4 years of training regarding for any statistical differences? In other words, are the reports more significant or less significant as the dental students advance to clinical confidence.

Reviewer #2: This study (survey) is well conducted and reported. Data in table 2 (Students' exposure to faculty of different genders) may be better presented as a figure (Histogram). Data in tables 3-6 should be defined e.g. Mean (SD)

Reviewer #3: 1. Discussion on the limitations of the manuscript should be expanded further. Comments regarding cultural/racial/ethnic differences are limited since the majority of the sample were Caucasian/White students. Lumping the non-White students together appeared to be for purely statistical analysis reasons, but can be misleading since there are likely differences in perception/experiences among African American/Black, Hispanic/Latino, Asian, and Mixed/multiple ethnic groups.

2. Given the limitations of the data collected (e.g., sample in terms of race/ethnicity is rather homogeneous), the statistical analysis appears appropriate; however, the authors should determine whether much can really be concluded about the non-White students.

3. Authors noted the following: "Yes - all data are fully available without restriction." "The data underlying the results presented in the study is available on request from Shaista Rashid."

4. The manuscript needs to be edited throughout for grammar and clarity. For example, the use of apostrophes, especially accurately discerning between singular and plural, needs to be addressed. On p. 2, please explain "assorted prospective." On p. 7, please clarify ". . . no difference between male and female students in preserving female faculty as role models." On p. 12, need to edit: "This difference could also by (sic) sex stereotyping of women as being. . ."

6. PLOS authors have the option to publish the peer review history of their article (what does this mean?). If published, this will include your full peer review and any attached files.

Reviewer #1: No

Reviewer #2: **Yes: **Mawlood Kowash

Reviewer #3: No

---

## [Author Response · Author response to Decision Letter 0]

12 May 2022

Dear Carlos Miguel Rios-González, 

Revision has been done according to suggestions

---

## [Decision Letter · Decision Letter 1]

15 Jun 2022

PONE-D-21-33779R1A Cross-sectional Study of Dental Students Perception of Dental Faculty Gender DifferencesPLOS ONE

Dear Dr. Shaista Rashid,

Thank you for submitting your manuscript to PLOS ONE. After careful consideration, we feel that it has merit but does not fully meet PLOS ONE’s publication criteria as it currently stands. Therefore, we invite you to submit a revised version of the manuscript that addresses the points raised during the review process.

We look forward to receiving your revised manuscript.

Kind regards,

Carlos Miguel Rios-González, Ph.D

Academic Editor

PLOS ONE

Reviewers' comments:

Reviewer's Responses to Questions

**Comments to the Author**

1. If the authors have adequately addressed your comments raised in a previous round of review and you feel that this manuscript is now acceptable for publication, you may indicate that here to bypass the “Comments to the Author” section, enter your conflict of interest statement in the “Confidential to Editor” section, and submit your "Accept" recommendation.

Reviewer #1: All comments have been addressed

Reviewer #2: All comments have been addressed

2. Is the manuscript technically sound, and do the data support the conclusions?

Reviewer #1: Yes

Reviewer #2: Yes

3. Has the statistical analysis been performed appropriately and rigorously? 

Reviewer #1: Yes

Reviewer #2: Yes

4. Have the authors made all data underlying the findings in their manuscript fully available?

Reviewer #1: Yes

Reviewer #2: Yes

5. Is the manuscript presented in an intelligible fashion and written in standard English?

Reviewer #1: Yes

Reviewer #2: Yes

6. Review Comments to the Author

Reviewer #1: The manuscript has improved with your additional edits. This reviewer has additional comments:

On page 4 regarding: Data were collected using a self-administered questionnaire adopted from the cognitive apprenticeship model in clinical practice.[18] Formatting...

Please check Formatting... and clarify the word.

On page 5 regarding: All questions have a not applicable (N/A) option.

Should it be All questions included a not applicable (N/A) option.

On page 13 regarding: This difference highlights the importance of gender matching and having females in academic roles to inspire female students.

Please rephrase this sentence by adding that: It to it is our belief or our reflection or that we speculate that "this difference highlights the importance of gender matching and having females in academic roles to inspire female students or something like that to highlight your reflection vs. a quotation."

It would have been instrumental to read the specific survey questions; however, the questionnaire was not included. The origin of the survey should be documented as well as where the questions came from or how they were formatted.

Reviewer #2: Table 4. Student’s mean (SD) perception scores for male and female faculty according to student’s gender and ethnicity

First row includes

Same applied for Male Faculty and Female Faculty twice and NO students Male and Female data.

or Table 6.

7. PLOS authors have the option to publish the peer review history of their article (what does this mean?). If published, this will include your full peer review and any attached files.

---

## [Author Response · Author response to Decision Letter 1]

28 Jun 2022

Reviewers comments are greatly appreciated. The changes have been made based on reviews comments. Please refer to "Response to Reviewers" letter for more detailed description.

---

## [Editor Report · Decision Letter 2]

5 Jul 2022

A Cross-sectional Study of Dental Students Perception of Dental Faculty Gender Differences

PONE-D-21-33779R2

Dear Dr. Shaista Rashid,

We’re pleased to inform you that your manuscript has been judged scientifically suitable for publication and will be formally accepted for publication once it meets all outstanding technical requirements.

Kind regards,

Carlos Miguel Rios-González, Ph.D

Academic Editor

PLOS ONE

---

## [Editor Report · Acceptance letter]

11 Jul 2022

PONE-D-21-33779R2 

A Cross-sectional Study of Dental Students Perception of Dental Faculty Gender Differences 

Dear Dr. Rashid:

I'm pleased to inform you that your manuscript has been deemed suitable for publication in PLOS ONE. Congratulations! Your manuscript is now with our production department. 

Kind regards, 

on behalf of

Dr. Carlos Miguel Rios-González 

Academic Editor

PLOS ONE